# Fine-Mapping Analysis of the Genes Associated with Pre-Harvest Sprouting Tolerance in Rice (*Oryza sativa* L.)

Seong-Gyu Jang [1,2], Backki Kim [1,2], Insoo Choi [1,2], Joohyun Lee [3], Tae-Ho Ham [4] and Soon-Wook Kwon [1,2,*]

1 Department of Plant Bioscience, Pusan National University, Miryang 50463, Republic of Korea
2 Life and Industry Convergence Research Institute, Pusan National University, Miryang 50463, Republic of Korea
3 Department of Crop Science, Konkuk University, Seoul 05029, Republic of Korea
4 Department of Agricultural Science, Korea National Open University, Seoul 03087, Republic of Korea
* Correspondence: swkwon@pusan.ac.kr; Tel.: +82-55-350-5506; Fax: +82-55-350-5509

**Abstract:** Pre-harvest sprouting (PHS) of rice (*Oryza sativa* L.) causes severe economic problems due to reduced grain quality and yield. Fine mapping was carried out to identify genes associated with PHS; the detected quantitative trait locus (QTL) was narrowed down to 50 Kbp using $F_{3:4}$ populations, four polymorphic insertion and deletion (InDel) markers, and two cleaved amplified polymorphic sequence (CAPS) markers. In one region, five candidate genes were detected, and the SNP and InDel in each gene (*Os01g0111400* and *Os01g0111600*) were confirmed to show the differences and resulting amino acid changes between parent plants. Based on haplotype, expression, and co-segregation analysis, the InDel in *Os01g0111600* was confirmed to be associated with the PHS trait. The results of this study could be applied to improve the PHS tolerance of *Japonica* rice varieties, and they also improved our understanding of the genetic basis underlying PHS tolerance.

**Keywords:** pre-harvest sprouting; whole-genome resequencing; fine-mapping; marker; *Oryza sativa* L.

## 1. Introduction

Rice (*Oryza sativa* L.) is the most important staple cereal crop for more than half of the world's population [1]. However, the security of the rice supply is threatened by climate change, including events such as typhoons, high temperatures, and other extreme weather conditions [2].

PHS is one of the most important adverse processes in rice because of the severe economic consequences of the markedly reduced grain quality and yield [3]. This occurs when seeds germinate during the maturing stage, before the harvest. Under normal seed dormancy (SD), maturation of the seed is arrested for various periods of time to allow it to germinate under favorable conditions. A lack of SD causes PHS in cereal crops [4]. In wild plant species, SD prevents germination of the panicle or grain under inopportune conditions. When matured rice panicles are not harvested in time or the harvested rice grains are not dried immediately after harvesting, rice cultivars with high PHS may exhibit germination when exposed to frequent rainfall and high temperatures [5–8]. SD is an adaptive trait that helps plants survive in harsh environments. However, PHS can break SD and promote seed germination under unfavorable conditions, which is also a crucial trait for plant survival [5,9].

Traits associated with seed germination, such as PHS, SD, and low-temperature germination (LTG), are highly complex, involving various physical and biochemical quality factors. They are quantitative traits subject to complex genetic control mechanisms, and germination-associated genes have been reported to be affected by auxin, abscisic acid (ABA), and gibberellin (GA) levels, which are major signaling molecules involved in germination induction [10].



Auxin is a phytohormone that is involved in various physiological processes, including phototropism, cell differentiation, cell expansion, floral opening, organ abscission, and seed germination [11]. Endospermic sugars are an essential energy source for seed germination and determine SD and germination by affecting ABA signaling [12]. GA promotes germination, which requires not only initiating embryo metabolic activity, but also breaking the physical barrier of the seed coat surrounding the embryo [13]. Several studies have proposed SD regulation as the predominant factor affecting PHS [4,14–18].

Many QTLs with these traits have been reported in different subspecies and ecotypes [15,19–21]. Fourteen QTLs related to SD were identified using recombinant inbred line (RIL) and restriction fragment length polymorphism markers [22], and *qSD-3*, *-5*, *-6*, and *-11* were detected using a double-haploid population [23]. *qPHS-11* was identified under field and greenhouse conditions through whole-genome resequencing [24]. *qSDR9.1* and *qSDR9.2* were identified on chromosome 9 using chromosome segment substitution lines and simple sequence repeat markers [25]. Only a few genes (*SD1-2* and *Sdr4*) have been identified through map-based cloning [26,27]. Using a genome-wide association study (GWAS), 10 loci associated with PHS were reported on chromosomes 1 and 4 based on a subset of SNPs, with 277 accessions from the 3000 Rice Genomes Project [28].

Improving PHS tolerance in rice is a major breeding objective [29]. To overcome PHS in rice under unpredictable weather conditions, genes and alleles associated with PHS must be identified. To address the incorporation of PHS tolerance into commercial rice cultivars, we conducted QTL and fine-mapping analysis with a mapping population derived from crossing high-quality *Japonica* varieties. We propose that the QTLs and candidate genes detected in this study could be used in commercial rice breeding programs.

## 2. Materials and Methods

### 2.1. Plant Materials

A PHS tolerance line (PHS-T) and a PHS susceptible line (PHS-S) were selected from a RIL population generated through crossing 'Jinsang', a parent producing high-quality rice, with 'Gopum', which also produces high-quality rice but is susceptible to PHS [30,31]. In the RIL population, parent plants were selected with similar agricultural traits, except for the PHS trait; as a result, the $F_2$ and $F_3$ populations showed pronounced segregation of tolerance and susceptibility, whereas traits such as plant height, heading date, culm length, and panicle length were uniform. In the previous study, 88 $F_2$ from a cross between PHS-T and PHS-S were used for QTL analysis to identify candidate regions [32]. The heterozygous plant was selected from the $F_2$ population to generate $F_3$ plants for further fine-mapping (Figure 1). PHS-T, PHS-S, and 241 $F_3$ plants were grown in the experimental field at Pusan National University in 2018 and 2020.

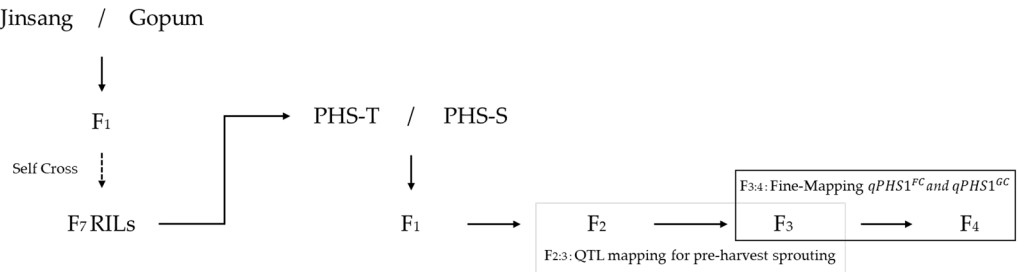

**Figure 1.** Process of developing the plant material in this study.

### 2.2. Phenotypic Evaluation of Pre-Harvest Sprouting

At forty-five days after flowering (DAF), three panicles from each parent and the 241 $F_3$ plants were evaluated for PHS rates under growth chamber (GC) conditions. Panicles were wrapped in paper towels and placed in water (100% relative humidity). Materials were incubated at 20 °C for seven days in a growth chamber [19]. For haplotype analysis, 93 Korean-bred varieties were also evaluated for PHS tolerance using 45 DAF panicles

under growth chamber conditions. All statistical analyses were conducted using R software version 4.1.3 for Windows.

### 2.3. Fine-Mapping of qPHS1$^{GC}$ and qPHS1$^{FC}$

To conduct a fine-mapping analysis, four InDel markers and two CAPS markers were designed from the variations between PHS-T and PHS-S at the detected QTL region reported in the previous study. Genotype data of PHS-T and PHS-S were acquired through whole-genome resequencing using an MGISEQ-2000 platform (MGI, Shenzhen, China) [32]. Genomic DNA was extracted from fresh rice leaves according to a modified CTAB protocol [33]. Polymerase chain reaction (PCR) was performed as follows: 95 °C for 5 min, followed by 35 cycles of 95 °C for 30 s, 55–58 °C for 30 s, 72 °C for 30 s, and a final elongation step at 72 °C for 7 min. PCR products were analyzed using a Fragment Analyzer TM (Agilent, Santa Clara, CA, USA).

### 2.4. RNA Extraction and qRT-PCR

RNA was isolated from 45 DAF rice grains of PHS-T and PHS-S using an RNeasy Plant Mini Kit (QIAGEN, Hilden, Germany), and samples were treated with RNase-Free DNase (QIAGEN, Hilden, Germany) to remove genomic DNA. Complementary DNA (cDNA) was synthesized using a SuperScript III Kit (Thermo Fisher Scientific, Boston, MA, USA), with primers for target genes designed using Primer 3 [34]. Reactions were performed in triplicate, and actin expression was used as an internal baseline control. The samples were kept on ice until immediately before qRT-PCR analysis using QuantStudio 1 (Thermo Fisher Scientific, Boston, MA, USA). The run parameters followed a standard PCR protocol, beginning with reverse transcription at 50 °C for 15 min, denaturation at 95 °C for 2 min, followed by 50 amplification cycles consisting of a denaturation step at 95 °C for 15 s and annealing and extension at 60 °C for 1 min, with fluorescence acquisition in the annealing/extension phase. QuantStudio Software version 1.5.2 (Thermo Fisher Scientific, Boston, MA, USA) was used to analyze the data. The cycle threshold was set to 0.05, and the baseline was chosen automatically. Two-tailed *t*-tests were performed using GraphPad software to compare samples. Statistical significance was reported at $p < 0.01$.

### 2.5. Genome Sequence Data for Korean-Bred Rice Varieties

A total of 93 Korean-bred rice varieties from the national agrobiodiversity center of the Rural Development Administration (RDA, Wanju, Republic of Korea) were used to detect variations of PHS (Table S1). Genomic data of the 93 varieties were produced with an approximately eight-fold mean coverage using an Illumina HiSeq 2500 Sequencing Systems Platform (Illumina Inc., San Diego, CA, USA). Raw reads were aligned against the rice reference genome (IRGSP 1.0) for genotype calling. To generate a genotype dataset, the following parameters were used for haplotype analysis: missing value < 1%, minor allele frequency > 5%, and heterozygosis ratio < 5%, as implemented in PLINK software [35].

### 2.6. Haplotype Analysis for Korean-Bred Varieties

SNPs and InDel regions were used for haplotype analysis, excluding missing and heterozygote regions. The average score and variety count were determined from phenotype data for each variety, and haplotypes that were significantly associated with the phenotype were identified. The online tool Gene Structure Display Server 2.0 was used for the visualization of gene structures and SNP and InDel positions [36].

### 2.7. Seed Germination Assay and ABA Treatment

To evaluate the effects of ABA on seed germination, grains were harvested at 45 DAF and were dehulled. To determine the ABA sensitivity regarding post-gemination growth, husked seeds were sterilized using 0.15% HgCl$_2$ and were washed in sterilized water. Then, the seeds were spread for germination on micro agar. Three days after incubation, seedlings with the same growth vigor were transferred onto untreated micro agar and 2

µM ABA-treated micro agar, respectively. Seeds and seedlings were grown in a growth chamber under a 14/10 h light/h dark cycle at 28 °C. Photos were taken, and shoot length was measured seven days after transplanting [37].

## 3. Results

### 3.1. Fine-Mapping of $qPHS1^{FC}$ and $qPHS1^{GC}$

In the previous study, the 17 linkage groups were constructed by filtering 376 SNPs from 12,737 SNPs between PHS-T and PHS-S. Under two different conditions, field condition (FC) and GC, two QTLs, i.e., $qPHS1^{FC}$ and $qPHS1^{GC}$, were detected in the same position under each condition, and this QTL was identified between ch01-0.040 and ch01-0.063 (Figure 2a) [32]. In this QTL, new polymorphic InDel markers and CAPS markers were developed for fine mapping of $qPHS1^{FC}$ and $qPHS1^{GC}$. To identify the specific genomic regions associated with PHS, the heterozygous $F_2$ plant was selfed to produce a subsequent $F_3$ population. A total of 241 $F_3$ plants were genotyped using InDel (In1–In4) and CAPS (SNP1 and SNP2) markers (Figure 2b and Table S2). Homozygous recombinants in the $F_3$ population were identified using markers and evaluated for PHS resistance. Based on the genotypes and tolerance phenotypes of the homozygous recombinants, a high-resolution map of the $qPHS1^{FC}$ and $qPHS1^{GC}$ locus was established. Through the procedure described above, a candidate region was detected and narrowed down that could be confined between markers SNP2 and In4 (Figure S1). Based on the results of genotype and phenotype assays, narrowed-down regions of $qPHS1^{FC}$ and $qPHS1^{GC}$ were found to be located in the interval between the SNP2 and In4markers, which spanned a 50 Kbp region in the genome sequence. This region contained five predicted genes (Figure 2c).

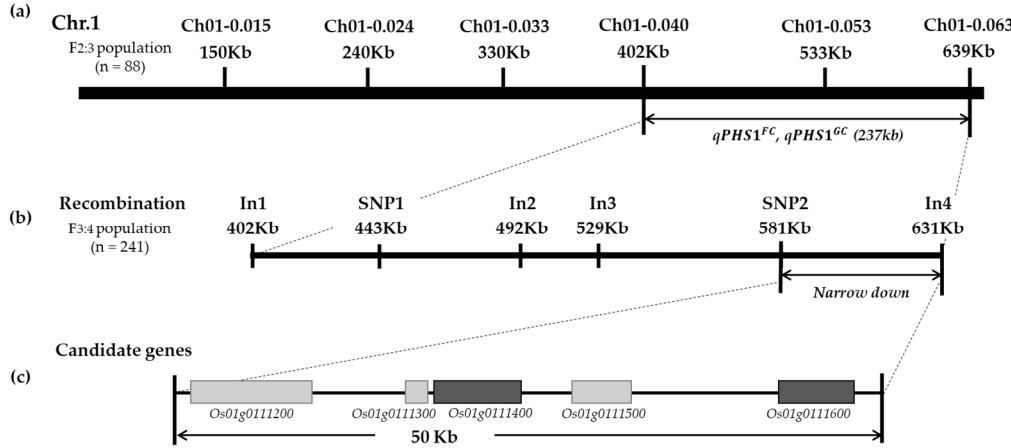

**Figure 2.** Fine-mapping of $qPHS1^{FC}$ and $qPHS1^{GC}$. (**a**) Physical map of the $qPHS1^{FC}$ and $qPHS1^{GC}$ locus using $F_{2:3}$ plants. (**b**) High-resolution mapping of the $qPHS1^{FC}$ and $qPHS1^{GC}$ locus using $F_{3:4}$ plants. (**c**) Candidate genes of $qPHS1^{FC}$ and $qPHS1^{GC}$. Black blocks indicate candidate genes with SNPs or InDel polymorphisms.

### 3.2. $qPHS1^{FC}$ and $qPHS1^{GC}$ Candidate Gene Prediction

Among five candidate genes of $qPHS1^{FC}$ and $qPHS1^{GC}$, three candidate genes, *Os01g0-111200*, *Os01g0111300*, and *Os01g0111500*, were monomorphic between PHS-S and PHS-T (Table 1). Therefore, the remaining two genes (*Os01g0111400* and *Os01g0111600*) were cloned and sequenced to detect the respective regions. *Os01g0111400* had one SNP between the parents. One SNP in *Os01g0111400* was detected in the third exon; the SNP was a T and an A in the PHS-T and PHS-S lines, respectively, and caused an amino acid change from serine to threonine (Figure 3a and Table 1). *Os01g0111600* showed a 16 bp deletion region in the fourth exon of PHS-S, giving rise to a stop codon-type of amino acid (Table 1 and Figure 3b).

**Table 1.** Candidate genes for the *qPHS1^{GC}* and *qPHS1^{FC}*.

| Gene Name | Length of CDS (bp) | Putative Function | Reported Gene | Sequence Variation |
|---|---|---|---|---|
| *Os01g0111200* | 1863 | Expressed protein | | No variation |
| *Os01g0111300* | 162 | Expressed protein | | No variation |
| *Os01g0111400* | 2106 | Transposon protein | | 1 SNP: T → A caused Ser → Thr |
| *Os01g0111500* | 888 | Regulation of root hair development | *OsbHLH125* *OsRSL1* | No variation |
| *Os01g0111600* | 525 | Regulation of ABA signaling-mediated seed germination | *OsMFT2* | 1 InDel: 16 bp deletion caused stop codon |

**(a)** *Os01g0111400*

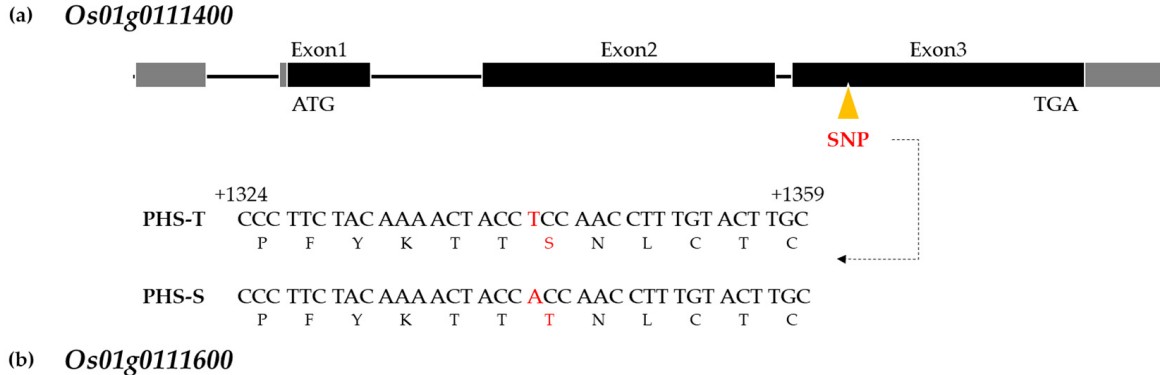

**(b)** *Os01g0111600*

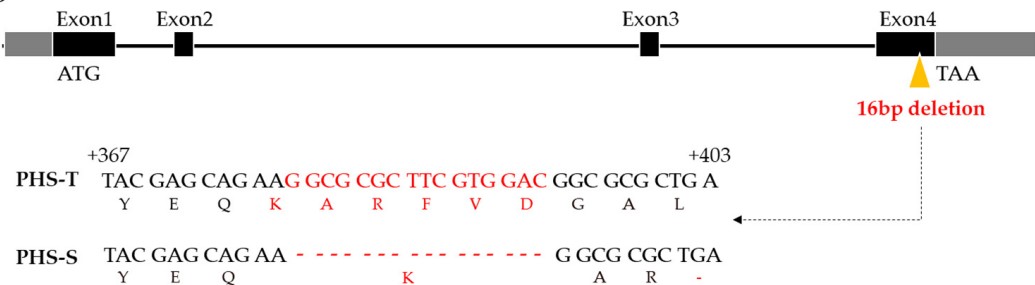

**Figure 3.** Gene structure and cDNA sequence comparison. (**a**) *Os01g0111400* structure and cDNA sequence comparison. (**b**) *Os01g0111600* structure and cDNA sequence comparison. The black and grey block and yellow vertical lines indicate exons, untranslated regions, and SNPs, respectively.

*3.3. Expression Patterns of Os01g0111400 and Os01g0111600*

Expression analysis was performed to investigate the transcript type of *Os01g0111400* and *Os01g0111600* in PHS-S and PHS-T during the grain-filling stage. For expression analysis, primers of target genes (*Os01g0111400* and *Os01g0111600*) were designed for qRT-PCR (Table S3). The relative expression level of *Os01g0111400* was similar between the two parents (Figure 4a), whereas the relative expression of *Os01g0111600* was markedly higher in PHS-T than in PHS-S, and the 16 bp deletion type exhibited reduced expression (Figure 4b). These results suggest that *Os01g0111600* acts as a positive regulator of PHS during seed germination rather than *Os01g0111400*, and the deletion region leads to PHS sensitivity.

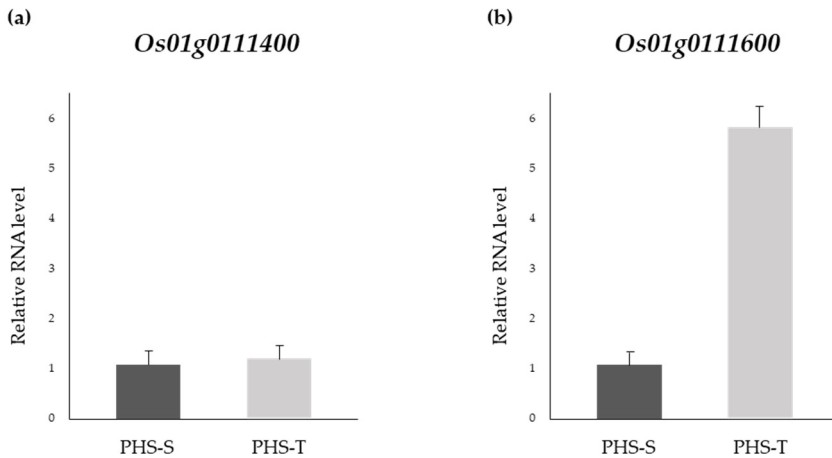

**Figure 4.** Expression analysis of candidate genes. (**a**) *Os01g0111400* expression in PHS-S and PHS-T 45 days after flowering. (**b**) *Os01g0111600* expression in PHS-S and PHS-T at 45 days after flowering. Values were normalized against the level of *OsAct1*. Error bars represent standard deviations (n = 3).

*3.4. Haplotype Analysis of Os01g0111600 in Korean-Bred Varieties*

Haplotype analysis was performed using genotype data of 92 Korean-bred varieties. The haplotype analysis of *Os01g0111400* separated three haplotypes, and the Korean-bred rice varieties with a haplotype with polymorphic SNP had a decreased PHS rate; however, it was not statistically significant (Figure S2). The other candidate gene, *Os01g0111600*, contained three SNPs in exon 1 and two SNPs and one InDel in exon 4 (Figure 5a). PHS-T and PHS-S were also confirmed to belong to Hap 2 and Hap 1 in *Os01g0111600*, respectively. Five SNPs and one InDel were divided into four haplotypes, with a maximum phenotypic variation of 95.6% for PHS between Hap 1, Hap 3, and Hap 4 (Figure 5b). In conclusion, InDel 1 in exon 4 caused amino acid changes, which showed significant differences regarding PHS. This gene was identified as a candidate gene for PHS tolerance in Korean-bred varieties (Figure 5b).

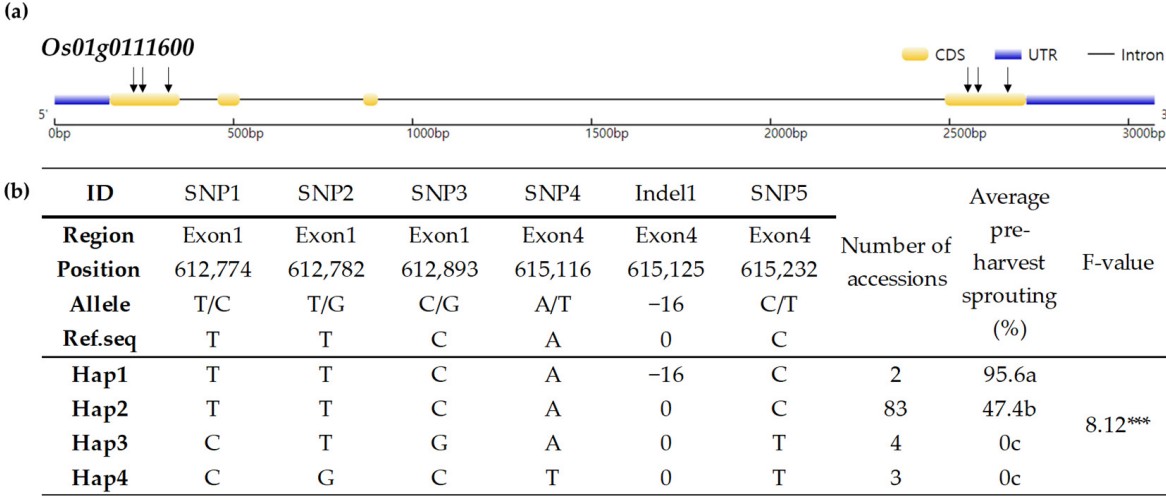

(a)

*Os01g0111600*

(b)

| ID | SNP1 | SNP2 | SNP3 | SNP4 | Indel1 | SNP5 | Number of accessions | Average pre-harvest sprouting (%) | F-value |
|---|---|---|---|---|---|---|---|---|---|
| **Region** | Exon1 | Exon1 | Exon1 | Exon4 | Exon4 | Exon4 | | | |
| **Position** | 612,774 | 612,782 | 612,893 | 615,116 | 615,125 | 615,232 | | | |
| **Allele** | T/C | T/G | C/G | A/T | −16 | C/T | | | |
| **Ref.seq** | T | T | C | A | 0 | C | | | |
| **Hap1** | T | T | C | A | −16 | C | 2 | 95.6a | |
| **Hap2** | T | T | C | A | 0 | C | 83 | 47.4b | 8.12*** |
| **Hap3** | C | T | G | A | 0 | T | 4 | 0c | |
| **Hap4** | C | G | C | T | 0 | T | 3 | 0c | |

**Figure 5.** Haplotype analysis of *Os01g0111600*. (**a**) Schematic representation of the gene structure and SNP positions in *Os01g0111600*. (**b**) Results of haplotype analysis of *Os01g0111600*. Yellow and blue blocks and gray lines indicate exons, untranslated regions, and intron regions, respectively. Black vertical bars represent SNPs and InDel regions. Hap: haplotype. Letters a, b, and c represent significant differences at *** $p < 0.001$ (Duncan's test).

### 3.5. Response to ABA Treatment of PHS-S and PHS-T

To investigate the effects of ABA on the germination performance of PHS-S and PHS-T, the germination rate under ABA treatment and normal conditions was evaluated using husked full seeds freshly harvested 45 DAF. Under ABA treatment conditions, seeds revealed some delay in germination; among them, the PHS-T showed higher sensitivity to ABA than PHS-S (Figure 6a). Shoot lengths of PHS-S and PHS-T controls (0 μM ABA) were 8.5 and 8.3 cm, respectively, whereas shoot lengths were 2.9 and 0.8 cm, respectively, with 2 μM ABA (Figure 6b). Thus, PHS-T exhibited some delay in germination under ABA treatment conditions, whereas PHS-S showed a relatively lower sensitivity as compared with PHS-T (Figure 6).

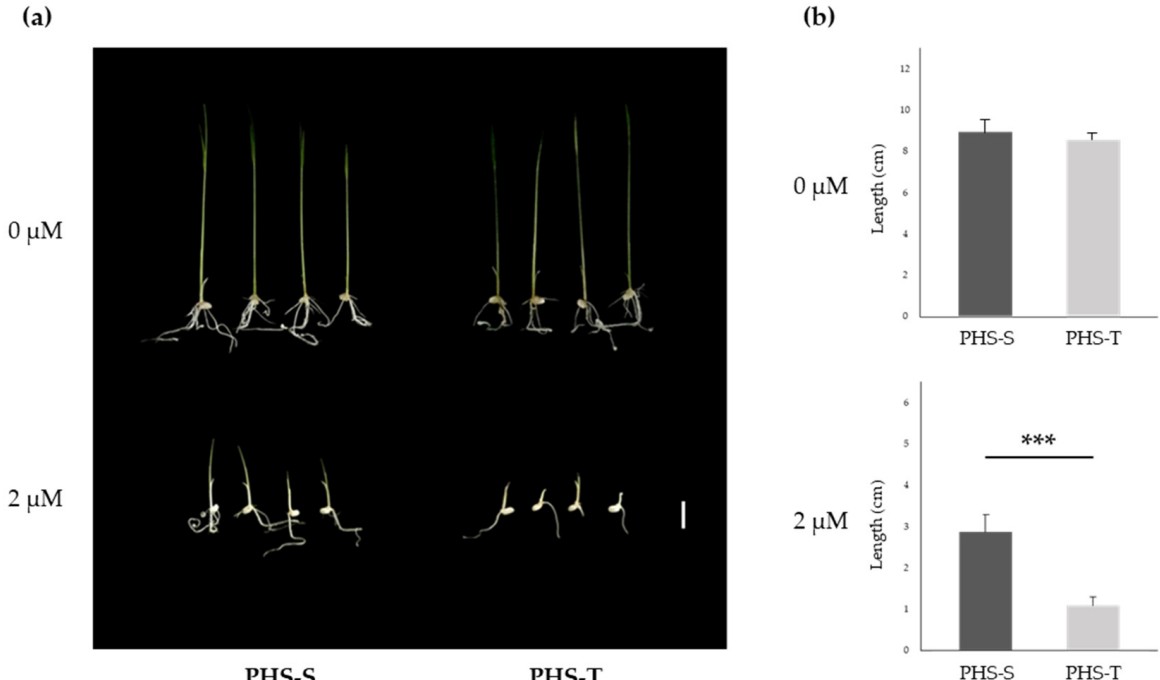

**Figure 6.** The response of PHS-S and PHS-T to ABA treatment during seed germination and post-germination stages. (**a**) Performance of PHS-T and PHS-S under untreated and 2 μM ABA treatment. Scale bars: 1 cm. (**b**) Shoot length of PHS-T and PHS-S under untreated and 2 μM ABA treatment. Error bars indicate standard deviations, n ≥ 10, *** $p < 0.001$ (Duncan's test).

### 3.6. Co-Segregation between Indel Markers of Os01g0111600 and the PHS Trait

Co-segregation analysis was conducted. The genotypes of the 16bp deletion region of *Os01g0111600* and the germination rate under PHS conditions were examined using panicles of $F_{2:3}$ and $F_{3:4}$ populations 45 DAF and a 16bp deletion detection marker (Table 2 and Figure 7a,b). The distribution of $F_{2:3}$ populations under growth chamber conditions showed that most plants were of the PHS-T type, and fewer were of the PHS-S type (Figure 7c). Among $F_{2:3}$ individuals under growth chamber conditions, the PHS-T type had a lower germination rate than the PHS-S type (Figure 7c). Additionally, $F_{2:3}$ populations under field conditions showed similar results to the growth chamber conditions (Figure 7d). The $F_{3:4}$ populations contained more PHS-T type than PHS-S type individuals, and the PHS-T type had a lower germination rate than the PHS-S type (Figure 7e).

**Table 2.** Primer used for co-segregation analysis of *Os01g0111600*.

| Marker Name | Type | Forward Primer (5′-3′) | Reverse Primer (5′-3′) | PHS-T-Type (bp) | PHS-S-Type (bp) |
|---|---|---|---|---|---|
| *Os01g0111600*-16bp | InDel | GGTGGGGATACACAGGTACG | CCTCTGGGAGTTGAAGTGGA | 166 | 150 |

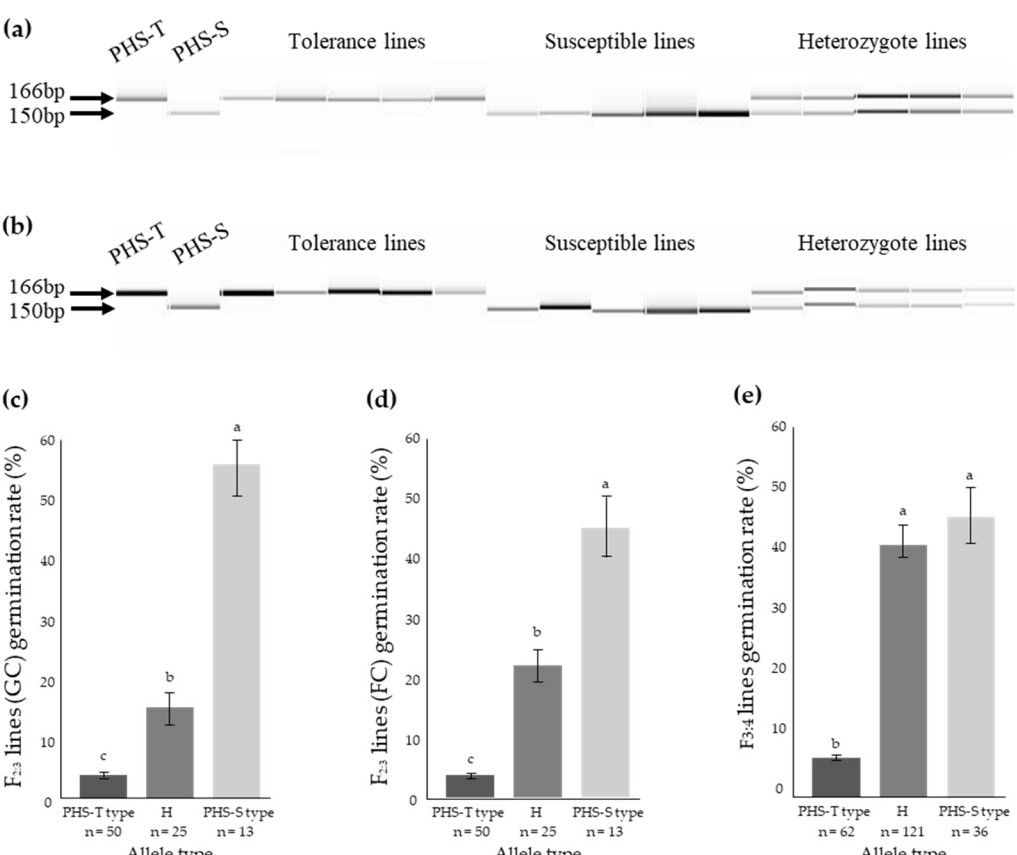

**Figure 7.** Co-segregation analysis of the *Os01g0111600*-16bp InDel marker using a fragment analyzer in F$_{2:3}$ and F$_{3:4}$ populations. Tolerance type: 166 bp; susceptible type: 150 bp. (**a**) Co-segregation analysis of the *OS01g0111600*-16bp InDel marker in the F$_{2:3}$ population. (**b**) Co-segregation analysis of the *OS01g0111600*-16bp InDel marker in the F$_{3:4}$ population. (**c**) Pre-harvest sprouting rate of the PHS-T type in the F$_{2:3}$ populations under growth chamber conditions. (**d**) Pre-harvest sprouting rate of the PHS-T type in the F$_{2:3}$ populations under field conditions. (**e**) Pre-harvest sprouting rate of the PHS-T type in the F$_{3:4}$ populations under growth chamber conditions. PHS-T type, H, PHS-T type, and n indicate the reference type, heterozygote, 16bp deletion type, and number, respectively. GC and FC represent growth chamber and field conditions, respectively. Letters a, b, and c represent significant differences at $p < 0.001$ (Duncan's test).

## 4. Discussion

PHS is one of the most important processes regarding rice yield and quality, and it is closely associated with SD, which inhibits seed germination under high humidity and temperature conditions [38]. Regarding traits associated with seed germination, the effects of PHS, SD, and LTG are extremely complex, involving various physical and biochemical factors. These are quantitative traits subject to complex genetic control mechanisms, and germination-associated genes have been reported to be affected by auxin, ABA, and GA levels, which are major signaling molecules involved in germination induction [10].

In order to identify candidate genes associated with the genetic regulation of PHS tolerance, we chose two inbred lines (PHS-T and PHS-S) from F$_7$ RIL and generated a population using PHS-T and PHS-S for analyses. The population in this study was derived from the same parental lines ('Jingsang' and 'Gopum'), and PHS-T and PHS-S were selected to have the same agronomy trait except for the PHS trait in the F$_7$ RIL population. Therefore, we used genotype resequencing because most nucleotide regions showed monomorphic SNPs and InDel regions in PHS-T and PHS-S. Through resequencing analysis, polymorphic SNPs were detected between PHS-T and PHS-S, and filtered SNPs were used to construct the genetic map for the QTL analysis in the previous study [32]. These materials were

derived to elaborate mapping and narrow down the QTL region from the fine-mapping analysis in this study.

*Os01g0111600* was reported to regulate ABA signaling genes and ABA levels in seed germination [39,40]. The Arabidopsis cytochrome P450 CYP707A gene codes enzymes involved in ABA catabolism. Overexpression of the CYP707A gene resulted in reduced ABA levels and decreased seed dormancy, while knockdown of the gene resulted in increased ABA levels and enhanced seed dormancy. Their findings suggest that the CYP707A gene plays a key role in ABA catabolism and seed dormancy regulation [39]. The NCED5 gene encodes an enzyme involved in ABA biosynthesis in rice seed dormancy. Overexpression of the NCED5 gene led to increased ABA levels and enhanced seed dormancy, while knockout of the gene resulted in decreased ABA levels and reduced seed dormancy. It was also suggested that other members of the NCED gene family also contributed to ABA accumulation and seed dormancy regulation in rice [40]. In a related study, OsMFT2 knockout lines exhibited pre-harvest sprouting, whereas OsMFT2 overexpression lines showed delayed germination [37]. ABA was demonstrated to be highly associated with seed germination, as well as low-temperature germination and dormancy, and the ABA level was determined to be related to delayed or exhibited germination [10,12]. To confirm the functions of candidate genes identified in this study, we evaluated the responses of ABA treatment in each parent plant. After broken seed dormancy of PHS-T and PHS-S, the growing degree of these seeds was compared in the normal micro agar condition and ABA treatment micro agar condition, respectively. PHS-S exhibited reduced ABA sensitivity in seed and post-germination, and PHS-T showed hypersensitivity to ABA in both germination steps, indicating the function of PHS-T in ABA signaling (Figure 6).

For co-segregation analysis, the *Os01g0111600*-16bp marker was developed to distinguish the 16bp InDel region in the *Os01g0111600* from the population in this study. PHS-S type varieties in the population showed an increasing rate of PHS, and PHS-T type varieties showed a decreased tendency compared to the PHS-S type (Figure 7). In the group of Korean-bred rice varieties, those classified as PHS-S types showed an increase in PHS rates (Figure 5). Therefore, using a developed marker from polymorphic InDel regions that affect variance is expected to help to identify PHS tolerance.

In this study, we identified candidate genes that are associated with PHS in the developed populations. A QTL region identified in a previous study was utilized to execute fine-mapping analysis, the *Os01g0111600* gene was considered the final candidate gene, and we inferred that a 16bp deletion was the core mutation site associated with PHS rate differences in the two *Japonica* parents. PHS leads to reduced rice grain quality and yield. We propose that the results of this study constitute an important resource for molecular breeding and for furthering the understanding of rice trait genetics.

## 5. Conclusions

In this study, PHS was surveyed using PHS-T, PHS-S, and developed populations to identify candidate genes associated with PHS tolerance. New polymorphic InDel markers and CAPS markers were developed for fine mapping, and five candidate genes significantly associated with PHS were identified on chromosome 1. Among these genes, in *Os01g0111600*, the 16 bp deletion in the fourth exon of PHS-S was detected; it resulted in a premature stop codon. Haplotype analysis revealed that InDel caused significant differences in PHS, and co-segregation analysis indicated that the 16 bp deletion region of *Os01g0111600* was significantly correlated with the PHS germination rate. We also observed that PHS-T exhibited higher sensitivity to ABA than PHS-S. Based on our findings, we provided a selection marker associated with PHS in *Os01g0111600*. This result will enable us to identify the specific genetic variants responsible for the observed differences and enhance our understanding of the genetic basis of the trait, ultimately aiding the development of improved crop yield and quality strategies.

**Supplementary Materials:** The following supporting information can be downloaded at https://www.mdpi.com/article/10.3390/agronomy13030818/s1. Figure S1. Recombination events and their effects on tolerance phenotypes in $qPHS1^{FC}$ and $qPHS1^{GC}$ genotypes. Figure S2: Haplotype analysis of candidate genes. (a) Schematic representation of the gene structure and SNP positions in *Os01g0111400*. (b) Results of haplotype analysis of *Os01g0111400*. Table S1: List of Korean-bred lines for this study. Table S2: Primers used for fine mapping of $qPHS1^{FC}$ and $qPHS1^{GC}$. Table S3: Primers used for qRT-PCR of candidate genes.

**Author Contributions:** Conceptualization, S.-W.K.; methodology, T.-H.H., J.L. and I.C.; validation, S.-G.J., B.K. and T.-H.H.; formal analysis, S.-G.J.; investigation, S.-G.J. and B.K.; resources, S.-W.K. and J.L.; data curation and writing—original draft preparation, S.-G.J.; writing—review and editing, B.K., J.L. and I.C.; supervision, project administration, and funding acquisition, S.-W.K. All authors have read and agreed to the published version of the manuscript.

**Funding:** This research was funded by the Ministry of Agriculture, Food and Rural Affairs (MAFRA), Grant Number 320105-3; and the National Research Foundation of Korea (NRF) funded by Ministry of Education, Science, and Technology, Grant Numbers 2021K1A3A1A61003041 and 2018R1D1A1B07051390.

**Institutional Review Board Statement:** Not applicable.

**Informed Consent Statement:** Not applicable.

**Data Availability Statement:** The data supporting the reported results can be found in the main text and the Supplementary Materials.

**Conflicts of Interest:** The authors declare no conflict of interest.

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
