# Peer review of "Fine-Mapping Analysis of the Genes Associated with Pre-Harvest Sprouting Tolerance in Rice (Oryza sativa L.)"

_agronomy, doi:10.3390/agronomy13030818_

Round 1

Reviewer 1 Report

In this manuscript, authors carried out fine-mapping a QTL for PHS in chromosome 1 narrowing down its location to a 50 kbp interval. Through expression and haplotype analysis of candidate genes, Os01g0111600 was confirmed to be associated with the PHS trait. Also, a selection marker for this gene was developed based on a 16 bp InDel. These results will be very much helpful in breeding PHS resistant rice varieties. However, it is needed to solve the following issues for publication of this manuscript.

Major points;

1. In Figure 2, it is needed to show data supporting the conclusion that the gene was narrowed down to SNP2-In4 interval. 

Minor points;  

1. In line 38-39, the meanings of the former phrase and latter phrase seem to same and too broad. Please improve this.

2. In line 86. “At” should be added to the beginning of the sentence.

3. In line 119 and 122, it would be better to change “Genomic data” to “Genome sequence data”.

4. In line 157, In6 was not shown in Figure2. Please check this.

5. In line 165 and 166, it would be better to change “Os01g0111200, Os01g0111300, and Os01g0111500, were identified with only monomorphic nucleotide regions between PHS-S and PHS-T (Table 1).” to “Os01g0111200, Os01g0111300, and Os01g0111500, were monomorphic between PHS-S and PHS-T (Table 1).”.

6. In Table 1. it would be better to change “Remarkable sequence” to “Sequence variation”, and “Non polymorphism” to “No variation”.

7. In line181, “markers” should be changed to “primers”.

8. In line 189, “at” should be added before “45 days”.

9. In line 216, the phrase of “PHS-S did not affect the ABA sensitivity of seedlings more than PHS-T.” is too much difficult to understand. Authors should solve this problem. 

10. In line 250. the word “quality” should be removed. 

11. In line 268, it would be better to change “confirmed” to “located”.

12. The sentence In line 296-298, “Among the Korean-bred rice varieties, PHS-S type varieties also had a PHS rate which tended to decrease (Figure 5). “, does not make sense. Please check this.   

13. The conclusion part needs to be improved. 

Author Response

Thanks to the reviewer's advice, the manuscript could be further improved.

Comments and Suggestions for Authors

In this manuscript, authors carried out fine-mapping a QTL for PHS in chromosome 1 narrowing down its location to a 50 kbp interval. Through expression and haplotype analysis of candidate genes, Os01g0111600 was confirmed to be associated with the PHS trait. Also, a selection marker for this gene was developed based on a 16 bp InDel. These results will be very much helpful in breeding PHS resistant rice varieties. However, it is needed to solve the following issues for publication of this manuscript.

Major points;

  1. In Figure 2, it is needed to show data supporting the conclusion that the gene was narrowed down to SNP2-In4 interval.
  • Thank you for your comment. For readers' comprehension, we have added fine-mapping contents to the "Materials and Methods" section and included a supplemental figure (Figure S1) in the "Results" section. (Line 96-98, 152-161)

Minor points;

  1. In line 38-39, the meanings of the former phrase and latter phrase seem to same and too broad. Please improve this.
  • Thank you for your comment. We have revised the corresponding sentence. (Line 39-42)

  1. In line 86. “At” should be added to the beginning of the sentence.
  • Thank you for your comment. We have revised based on your recommendations. (Line 88)

  1. In line 119 and 122, it would be better to change “Genomic data” to “Genome sequence data”.

  • Thank you for your comment. We have revised based on your recommendations. (Line 121)

  1. In line 157, In6 was not shown in Figure2. Please check this.
  • Thank you for your comment. We erroneously indicated In6. it should have been In4.
    We correct it rightly. (Line 160)

  1. In line 165 and 166, it would be better to change “Os01g0111200, Os01g0111300, and Os01g0111500, were identified with only monomorphic nucleotide regions between PHS-S and PHS-T (Table 1).” to “Os01g0111200, Os01g0111300, and Os01g0111500, were monomorphic between PHS-S and PHS-T (Table 1).”.
  • Thank you for your comment. We have made the necessary revisions based on your recommendations. (Line 170-171)

  1. In Table 1. it would be better to change “Remarkable sequence” to “Sequence variation”, and “Non polymorphism” to “No variation”.
  • Thank you for your comment. We have made the necessary revisions based on your recommendations, including modifications to the content of Table 1.

  1. In line181, “markers” should be changed to “primers”.
  • Thank you for your comment. We have revised based on your recommendations. (Line 187)

  1. In line 189, “at” should be added before “45 days”.
  • Thank you for your comment. We have revised m based on your recommendations. (Line 195)

  1. In line 216, the phrase of “PHS-S did not affect the ABA sensitivity of seedlings more than PHS-T.” is too much difficult to understand. Authors should solve this problem.
  • Thank you for your comment. We have modified the content to ensure that it can be understood by the readers. (Line 222-224)
  1. In line 250. the word “quality” should be removed.
  • Thank you for your comment. We have made the necessary revisions based on your recommendations. (Line 255-256)

  1. In line 268, it would be better to change “confirmed” to “located”.
  • Thank you for your comment. Due to the suggestion of another reviewer, we have removed the sentence containing the word from the manuscript.

  1. The sentence In line 296-298, “Among the Korean-bred rice varieties, PHS-S type varieties also had a PHS rate which tended to decrease (Figure 5). “, does not make sense. Please check this.
  • Thank you for your comment. We have modified the content to ensure that it can be understood by the readers. (Line 294-296)

  1. The conclusion part needs to be improved.
  • Thank you for your comment. We have made revisions to the "Conclusion" section in order to improve it. (Line 305-318)

Reviewer 2 Report

1.     This work described the fine mapping of a loci associated with rice pre-harvest sprouting, it is interesting on rice breeding. The manuscript could be considered for publication after moderate revision and English improvement.

2.     In section “materials and method”, please describe the definition of “F2:3” and “F3:4” more clearly to avoid confuse. For example, in my understand, in F2 population, 88 F2 individual plants were used for mapping investigation, the leaves from F2 plants were sample for genotype, and the panicles (with F3 seeds) from F2 plants were used to germinate for phenotype. If so, some describe should be revised:

In lines 86-87, “the 241 F3:4 population” might be describe as: “the F3:4 population”, or “the 241 F3 plants and their corresponding panicles with F4 seeds”

In line 88, “Three panicles of each line” might be describe as “Three panicles of each plant”, or ” Three panicles of each F3 plant” or “Three panicles with F4 seeds from each F3 plant”

3.     In the section of "materials and methods", more details of fine mapping methods is need, and in the section of "results", information and details of recombinants is need to show readers how to define the chromosome region where candidate genes are located.

4.     In lines 151-152, the statement of “the F2:3 populations were selfed to produce a subsequent generation”, might be revised as: “the heterozygous F2 plants were selfed to produce a subsequent F3 population”.

5.     In lines 152, ” 241 F3:4 plants” Might be revised as:  ” 241 F3 plants”.

6.     In line 161, the authors describe: “Black and white blocks indicate candidate genes”, and in lines161-162, the authors describe: ” Black blocks indicate candidate SNPs or InDel regions.” These two statements might make readers confuse. It might be revised as: “Black blocks indicate candidate genes with SNPs or InDel polymorphisms.”

7.     Lines 216-217, the describe of “but PHS-S did not affect the ABA sensitivity of seedlings more than PHS-T.” should be revised.

8.     Line 219, the statement of “Response of ABA treatment to PHS-S and PHS-T.” should be revised.

9.     Lines 219-220, the statement of “ (a) ABA treatment of PHS-S and PHS-T at the stages of seed germination and post-germination growth.” Should be revised.

10. There is no need to repeat the results in the "discussion" section, so there is no need for some content, such as lines 254-280 and 300-308.

11. Lines 281-282, the statement of “Os01g0111600 was reported to positively regulate ABA-responsive genes by interacting and negatively regulating seed germination in rice”should be considered for revision.

Author Response

Thanks to the reviewer's advice, the manuscript could be further improved.

  1. This work described the fine mapping of a loci associated with rice pre-harvest sprouting, it is interesting on rice breeding. The manuscript could be considered for publication after moderate revision and English improvement.
  • Thank you for your comment. We appreciate your sincere review and valuable advice to improve the manuscript. This manuscript has been edited the English by MDPI's English editing service (English Editing ID : english-60906). We check it again and corrected something, and hope that the manuscript has improved compared to before.

  1. In section “materials and method”, please describe the definition of “F2:3” and “F3:4” more clearly to avoid confuse. For example, in my understand, in F2 population, 88 F2 individual plants were used for mapping investigation, the leaves from F2 plants were sample for genotype, and the panicles (with F3 seeds) from F2 plants were used to germinate for phenotype. If so, some describe should be revised.
  • Thank you for your comment. Your understanding of our content is correct. We have made the necessary revisions to the "Materials and Methods" section based on your recommendations. (Line 80-83)

In lines 86-87, “the 241 F3:4 population” might be describe as: “the F3:4 population”, or “the 241 F3 plants and their corresponding panicles with F4 seeds”

  • We have made the necessary revisions based on your recommendations. (Line 88)

In line 88, “Three panicles of each line” might be describe as “Three panicles of each plant”, or ” Three panicles of each F3 plant” or “Three panicles with F4 seeds from each F3 plant”

  • We have made the necessary revisions based on your recommendations. (Line 89)

  1. In the section of "materials and methods", more details of fine mapping methods is need, and in the section of "results", information and details of recombinants is need to show readers how to define the chromosome region where candidate genes are located.
  • Thank you for your comment. For readers' comprehension, we have added fine-mapping contents to the "Materials and Methods" section and included a supplemental figure (Figure S1) in the "Results" section. (Line 96-98, 152-161)

  1. In lines 151-152, the statement of “the F2:3 populations were selfed to produce a subsequent generation”, might be revised as: “the heterozygous F2 plants were selfed to produce a subsequent F3 population”.
  • Thank you for your comment. We have made the necessary revisions based on your recommendations. (Line 153-154).

  1. In lines 152, ” 241 F3:4 plants” Might be revised as: ” 241 F3 plants”.
  • Thank you for your comment. We have made the necessary revisions based on your recommendations. (Line 154)

  1. In line 161, the authors describe: “Black and white blocks indicate candidate genes”, and in lines161-162, the authors describe: ” Black blocks indicate candidate SNPs or InDel regions.” These two statements might make readers confuse. It might be revised as: “Black blocks indicate candidate genes with SNPs or InDel polymorphisms.”
  • Thank you for your comment. We have made the necessary revisions based on your recommendations. (Line 166-167)

  1. Lines 216-217, the describe of “but PHS-S did not affect the ABA sensitivity of seedlings more than PHS-T.” should be revised.
  • Thank you for your comment. We have revised the corresponding sentence. (Line 222-224)

  1. Line 219, the statement of “Response of ABA treatment to PHS-S and PHS-T.” should be revised.
  • Thank you for your comment. We have revised the corresponding sentence. (Line 225-226)

  1. Lines 219-220, the statement of “ (a) ABA treatment of PHS-S and PHS-T at the stages of seed germination and post-germination growth.” Should be revised.
  • Thank you for your comment. We have revised the corresponding sentence. (Line 226-227)

  1. There is no need to repeat the results in the "discussion" section, so there is no need for some content, such as lines 254-280 and 300-308.
  • Thank you for your comment. We have removed the redundant contents in the "Discussion" section that overlapped with the content in the "Results" section.

  1. Lines 281-282, the statement of “Os01g0111600 was reported to positively regulate ABA-responsive genes by interacting and negatively regulating seed germination in rice”should be considered for revision.
  • Thank you for your comment. We have revised the corresponding sentence. (Line 270-271)
